# Pollution Source Localization in Wastewater Networks

**DOI:** 10.3390/s21030826

**Published:** 2021-01-26

**Authors:** Krystian Chachuła, Robert Nowak, Fernando Solano

**Affiliations:** 1Faculty of Electronics and Information Technology, Warsaw University of Technology, 00-665 Warsaw, Poland; k.chachula@tele.pw.edu.pl (K.C.); fs@tele.pw.edu.pl (F.S.); 2Blue Technologies, 02-684 Warsaw, Poland

**Keywords:** continuous monitoring, information fusion and sensors, internet of things, multisensor fusion

## Abstract

In December 2016, the wastewater treatment plant of Baarle-Nassau, Netherlands, failed. The failure was caused by the illegal disposal of high volumes of acidic waste into the sewer network. Repairs cost between 80,000 and 100,000 EUR. A continuous monitoring system of a utility network such as this one would help to determine the causes of such pollution and could mitigate or reduce the impact of these kinds of events in the future. We have designed and tested a data fusion system that transforms the time-series of sensor measurements into an array of source-localized discharge events. The data fusion system performs this transformation as follows. First, the time-series of sensor measurements are resampled and converted to sensor observations in a unified discrete time domain. Second, sensor observations are mapped to pollutant detections that indicate the amount of specific pollutants according to a priori knowledge. Third, pollutant detections are used for inferring the propagation of the discharged pollutant downstream of the sewage network to account for missing sensor observations. Fourth, pollutant detections and inferred sensor observations are clustered to form tracks. Finally, tracks are processed and propagated upstream to form the final list of probable events. A set of experiments was performed using a modified variant of the EPANET Example Network 2. Results of our experiments show that the proposed system can narrow down the source of pollution to seven or fewer nodes, depending on the number of sensors, while processing approximately 100 sensor observations per second. Having considered the results, such a system could provide meaningful information about pollution events in utility networks.

## 1. Introduction

In recent years, there has been a growing global concern regarding the security of water distribution systems (WDSs) and wastewater networks (WWNs). WDSs and WWNs are spatially diversified, pervasive, and linked to the basic needs of human society. They are therefore considered as critical infrastructures by all national security agencies.

Events occurring in these systems that can have an impact on civilians include the following:
Accidental contamination of a WDS leading to contamination as a result of non-potable water surrounding pipe breaks and leaks, or from the back-flow of polluted water from customer facilities.Intentional contamination of WDSs by terrorists, i.e., the deliberate poisoning of a given population downstream.Prohibited connections to storm water networks that could potentially cause pollution of natural water bodies.Careless dumping of waste over WWNs, which could lead to explosions and cause major catastrophes due to the constant presence of flammable gasses produced by existing bacteria.Discharge of toxic substances into a WWN, which may result in the release of illegal and harmful concentrations of pollution into the environment.

### 1.1. Case Studies

There have already been cases of intentional contamination of WDSs by terrorist. Water-related terrorist activities have been reported in ancient Rome, in the United States during the Civil War, in Europe and Asia during World War II, and during the Kosovo conflict of 1999 [1].

With regard to WWNs, the discharge of sulfuric acid (H_2_SO_4_) to sewers could originate from applications, such as etching of semiconductors, accumulator acid or the production of organic chemical substances [2]. Sodium hydroxide (NaOH) is widely used for cleaning surfaces in metal processing in industrial applications [3], whereas discharge of sodium sulfate (Na_2_SO_4_) can be caused by the regeneration of cation exchange resins, which are used for softening water in industrial water treatment [4]. Illegal discharge of such dangerous harsh industrial waste into sewage networks could be harmful for the biological stage of waste water treatment plants (WWTPs), its personnel, sewer pipes, and the general public. Once of the most recent cases of this occurred in December 2016, when the WWTP of Baarle-Nassau, Netherlands, failed [5]. The operator noticed that the biological treatment stage failed completely as the pH level in the aeration tank was extremely acidic, with a pH level of nearly 1. This damage was caused by the improper disposal of large volumes of wastewater containing high concentrations of sulfuric acid into the sewage system.

### 1.2. Past Works

During the past decade, to mitigate the effects of potential polluting events in water systems, the research and industrial communities have focused primarily on three lines of research and development: (1) innovative sensor technologies for monitoring pollutant levels; (2) network planning solutions aimed at providing optimal network coverage constrained to a given capital, expenditure or event likelihood; and (3) source localization methods for detecting the most likely injection point of a pollutant if such an event occurs.

In recent years, several project initiatives [6,7] and prototypes of sensor systems [8,9,10,11,12,13,14,15,16,17,18,19,20,21,22] for wastewater monitoring have been proposed and studied. These include the design of sensors (electrochemical sensors, optical sensors, mass spectrometry, ion spectrometry, etc.) for manholes, main sewer lines, water bodies, and basins at the WWTP for estimating the presence or concentration of specific pollutants at the point and time of measurement. These systems are not capable of inferring the localization in the network where the pollutant was introduced.

A second line of research involves the design of planning methods for the deployment of a set of sensors in a given network [1,23,24,25,26,27] so that the arrangement of the deployed sensors maximizes the likelihood of detecting any anomaly. Furthermore, some research was done on the topic of portioning WDSs. Di Nardo et al. in [28] proposed a methodology that combines an algorithm for the automated creation of district metered area (DMA) boundaries with practical criteria for DMA design. Ciaponi et al. in [29] focused on proving the benefits of partitioning by simulating a discharge of cyanide and investigating the influence of district isolation on the security of a water supply system.

In this article, we focus on the third line of research: localization methods for detecting the most likely injection point of a pollutant. We present and evaluate a data fusion framework that aids the localization of the most likely source of pollution for sewer networks. The data fusion framework processes measurements collected by point-detection sensors in the sewage network (as input) and it estimates (as output) the likelihood that a sewage network inlet was the source of the pollutant.

In 2008, Di Cristo and Leopardi proposed an iterative procedure for identifying the source of pollution among a set of nodes that are monitored by sensors in a WDS [30]. Di Cristo and Leopardi identified the most likely source of the pollution by solving an optimization problem. The problem formulation minimizes the squared difference between the values measured by a sensor and the hydraulic model values for each node, where the hydraulic conditions of the network allowed for pollution. However, Di Cristo and Leop-ardi did not consider the localization of the pollution source outside the set of monitored nodes.

In the same year, Preis and Ostfeld proposed a genetic algorithm (GA) for solving a similar optimization problem [31]. However, the objective function was formulated as the least-squares difference between the detected (at the monitoring stations) and simulated contaminated values. The GA evaluated different permutations of four problem variables: (1) the contaminant injection node (integer), (2) the injection start time (real), (3) the injection duration (real), and (4) the injection mass rate (real). Two additional studies considered the usage of a GA to solve similar objective functions with the same four problem variables. In [32], the minimization of the absolute value of the difference between the values measured by a sensor and the hydraulic model values was proposed. A year later the minimization of the normalized square difference between simulated and measured contaminant concentration values was depicted [8].

In 2009, Huang and McBean provided provided a heuristic solution to the problem using a different approach [33]. Huang and McBean assumed that the insertion time of a pollutant into the network was known then, the heuristic determined whether a measurement corresponded with the insertion of the pollutant at the source by comparing the arrival times of the measurement to a monitored node with the expected arrival time window estimated by a hydraulic model. By considering a sequence of measurements over all nodes in the network, Huang and McBean estimated the probability that such an injection event was caused at a given node.

In 2010, Sanctis et al. presented the contamination status algorithm (CSA), which is based on the particle backtracking algorithm (PBA) [34]. The PBA infers the mass concentration ratio that every output node in a network shall receive over time from any of its upstream (input) nodes as a linear function. The CSA categorizes the state of the input nodes as safe, unsafe, or unknown based on the concentration ratios over all feasible input–output pairs of nodes in a network derived by the PBA.

All previously mentioned studies provide a methodology for localizing the source of a pollutant injection in a WDS. To the best of our knowledge, the work presented by the authors in this article is the first one inferring on the localization of a pollutant injection in sewage WWN.

In addition to the anomaly localization problem in WDSs and WWNs, there is the anomaly detection problem—the source of an anomaly cannot be found even if an abnormal time series of measurements occurs. Support vector machine (SVM) approaches for anomaly detection are widely used [35,36,37,38,39], but these are not effective at detecting a gradual anomalous change of sensor values in a time-series [35]. Numerous studies have used artificial neural networks (ANNs) for anomaly detection [35,38,40,41,42,43].

In the present study, we propose an algorithmic solution that assumes the following:
the network topology is known and static,the localization of the sensor devices is known and static,the number of sensor devices is limited and not all points of the sewage network are monitored,the sensor devices have heterogeneous but complementary sensing capabilities, andthe sensor devices sample water quality at a subset of network junctions at arbitrary sampling times.

This article is organized as follows. Section 2 describes the data fusion strategy, Section 3 presents the process and the outcomes of its evaluation, and Section 4 contains the conclusion of our findings.

## 2. Methods

The sewage network is represented by a directed acyclic graph G(V,E). A node v∈V represents a sewage network junction or spot, such as a building or sewage well. One or more sensors could be deployed in a node. Each edge e∈E represents a pipe between two nodes. The attribute $o_{e}$ of every edge *e* provides the current flow propagation time offset (lag) it introduced between its two connecting nodes. The direction of an edge corresponds to the direction of the wastewater flow. Additionally, it is assumed that (1) the graph *G* is consistent, (2) each node is connected to the root by exactly one path, and (3) the graph *G* contains a node representing the sink (drain). The sewage network’s sink is the location where all the wastewater exits the network. Hereinafter, we use the terms “sink” and “root” interchangeably. The graph *G* is a directed tree created under the assumptions mentioned above. Section 3 shows two examples of such networks, where root nodes are marked as “1”.

The sensors provide measurements of the wastewater properties in the form of observations *O* [44]:(1)O=〈Q,v,t,y,Δy〉,where

*Q* is the entity, *v* is the spatial location of the measurement, *t* is the time-stamp of the measurement, *y* is a digital representation of the measured value, and Δy is the uncertainty. Possible entity values for *Q* include electrical conductance, and pH and concentration of a specific compound. The spatial location of an observation *O* corresponds to the node *v* in *G* where the measurement was taken.

We define the vector of all observed entities as $Q=\langle Q_{1},Q_{2},\cdots ,Q_{N}\rangle$.

In the presented system a finite list of substances (compounds) to be tracked is represented by set $C=\{C_{1},C_{2},C_{3},\cdots ,C_{M}\}$, where $C_{i}\in C$ is a compound. Predefined functions are used to convert measured values to amounts for each compound $C_{i}$. It should be noted that C included not only pollutants, but also other compounds, primarily those that are generally present in wastewater.

The data fusion algorithm consists of five steps: resampling, pollution quantification, downstream propagation, tracking, and event generation (Figure 1). These steps are repeated. The input data for resampling (first step of the algorithm) consists of sensor measurements, the output data of the resampling is the input for the pollution quantification, etc.

### 2.1. Resampling

Each sensor in the system is capable of sampling at a different time period. In the resampling step, we convert sensor measurements into sensor observations in a unified discrete-time domain by setting a common sampling time period for all sensors and estimating the value of sensor measurements that were not initially collected. Therefore, for each iteration of the data fusion algorithm, the values of *y* and Δy are calculated. This process utilizes linear interpolation when the sampling period *T* is greater than the sensor measurement period or if there are missing measurements, and mean aggregation, when the sampling period is less than the sensor measurement period.

Sensor observations resulting from this step are represented as indicated below.
(2)$$O^{\prime }=\langle Q,v,k,y,\Delta y\rangle ,\;\mathrm{where}$$
*k* is the discrete-time step k=0,1,2,⋯.

Time steps are referenced to a fixed point in time $t_{0}$ so that measurements taken at $t_{0}$ have k=0. In the present study, we assume uniform time sampling. Therefore, discrete-time step *k* represents $t_{k}=t_{0}+k\cdot T$.

### 2.2. Pollutant Quantification

The pollution quantification step converts sensor observations $O^{\prime }=\langle Q,v,k,y,\Delta y\rangle$ into the identification and quantification of sought compounds.

The pollution quantification step yields a set of *pollution detections*
$D=\langle D^{1},D^{2},\cdots ,D^{M}\rangle$. Each pollution detection $D^{i}$ takes the form $D^{i}=\langle C_{i},v,k,a^{i},\Delta a^{i}\rangle$, where $a^{i}$ is the amount in liters of a substance $C_{i}$ that is detected with uncertainty $\Delta a^{i}$ by node *v*.

Pollution detections are created using the following method. For each sensor observation $O^{\prime }$, every compound C∈C is considered independently. A potential discharge amount is calculated using the mapping function f(C,y,Δy)→〈C,a,Δa〉, where *y* is the measured value of entity *Q*, *C* is the compound, and *a* is the amount.

A threshold value σ is considered for filtering out sensor observations that are below the noise level. In other words, only if the inferred pollution detection amount $a^{i}$ is greater than the threshold σ, pollution detection is created and added to the detection set. The algorithm used to calculate pollution detections is depicted in Algorithm 1.
**Algorithm 1:** Pollution quantification algorithm
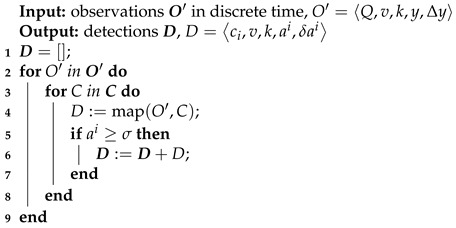



The $map(O^{\prime },O)$ function in line 4 of Algorithm 1 compound amount *a* from an input sensor observation *y* in the following way. Let $z=\left|y-b\right|$, where *y* is a sensor observation value and *b* is the baseline, which is defined as the sensor observation value when no compound is present in the proximity of the sensor. In the presented study, we assume linear mapping from *z* to the compound amount, a=αz. Parameter α specifies how a unit amount of a compound can be quantified into pollutant volume units.

A new detection object is created only if amount *a* exceeds the detection threshold. Thresholds are set per compound and are constant in time. These thresholds allow us to filter out insignificant detected amounts caused by small fluctuations of measured values.

### 2.3. Downstream Propagation

The downstream propagation step infers additional pollution detections in vertices of the graph where no sensors are installed. The majority of vertices are like this as we assume the number of available sensors to be limited, due to either high capital or operational costs.

The inferred time of arrival of a pollutant is generated from pollution detections D using Algorithm 2.

A depth-first search algorithm is used to create new detections downstream from their original nodes with maximum depth *d*, for each pollution detection D∈D.

New pollution detections are inferred by considering the propagation model of compounds in the utility network between neighbor vertices. For this, we consider the following three simplifications.

The propagation time of a substance for an edge, $o_{e}$, is known, constant in time, and equal for every compound. In practice, this condition is satisfied only when the flow characteristics do not change in time and the flow rate for each compound is the same.The total amount of a discharged compound does not change as the substance flows through the network. In practice, a substance may either react with other domestic waste and change its intrinsic characteristics, or may adhere to the sewage pipe walls.The sensors have infinite resolution and no noise. Therefore, tiny volumes of diluted compounds in the network over time can be measured.

**Algorithm 2:** Detection propagation algorithm

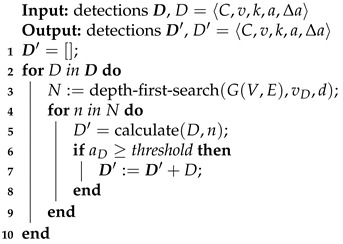



Inferred detections $D^{\prime }$ with amounts less than a given threshold (representing the process noise) are not considered. After detection propagation, pollution detections are associated with almost every vertex in the graph.

### 2.4. Tracking

The tracking algorithm clusters pollution detections by the detected compound. A cluster of pollution detections associated with a detected compound is named a *track*. Therefore, pollution detections can not be associated with a track if there is a difference in compounds between the detection and the track.

In this article, a Kalman filter is used for predicting the most probable location of a detected compound within the network in a previous algorithm iteration (for time $t_{i-1}$). The tracking algorithm updates the most probable location for time $t_{i}$ using the pollution detections calculated in the previous two steps.

The filter state (Equation (Equation 3)) represents the location in the network of a substance at a given point in time. For each track, the most probable amount *a* of a compound, as well as the most probable location *d* (as a function of time), are determined. Location is expressed as a real number equal to the distance from the network sink.
(3)$${\hat{x}}_{k|k}={\left[\begin{array}{cc} a & d \end{array}\right]}^{T}$$

The precise location of a compound within a track can be calculated at any time based on the fact that only one path connects each node to the sink and that the starting node of the track is stored. This localization scheme places a compound on the graph edge located at vector 〈u,v,α〉, where u,v are the source and the destination of the edge, respectively, and $\alpha \in \left[0,1\right)$ is a number describing the position relative to the edge.

The tracking algorithm (Algorithm 3) assigns pollution detections to tracks, creates new tracks, and removes stale tracks. Once the location of compounds within tracks are predicted by the Kalman filter, an assessment of whether the pollution detection can be supported is performed by comparing the amounts $a_{d}$, $a_{t}$ and graph distance $D_{g}$ between the detections and the track representatives. If these values are less than their respective thresholds, the detection is counted as supporting the track. If the detection cannot be associated with the existing tracks, a new track is created. Tracks with no new associations over several previous algorithm iterations are labeled as outdated and are removed.
**Algorithm 3:** Tracking algorithm
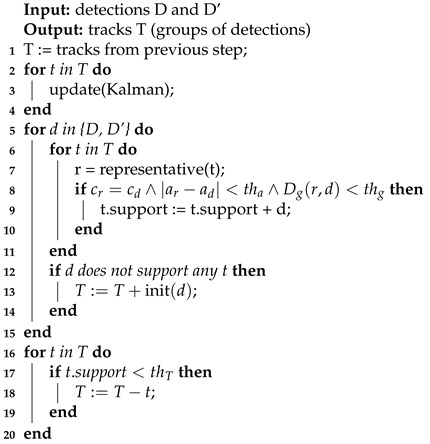



### 2.5. Event Generation

The final step, namely, *event generation*, only considers tracks that have a large number of supporting (associated) detections. Events are created for each of these tracks, where an event represents the discharge of a compound into a node of the graph.

Event generation is depicted in Algorithm 4.
**Algorithm 4:** Event generation algorithm
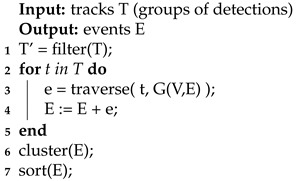



Possible events are generated for each important track. Subsequently, equivalent events from different paths are transformed to a single event with the confidence being equal to the sum of the confidences of those events and the compound amount being equal to the maximum of the amounts in a cluster. Finally, the events are sorted in descending order of confidence.

### 2.6. Implementation

The data fusion algorithm was implemented as a Python package with a modular layout. This enabled the user to replace any of the modules with ones that were more suited to their specific use. This was especially important as the subsequent modules of the system were then simplified compared to the real world. The sampling process relied on linear interpolation and mean aggregation. The amounts of the compounds were assumed to be in a linear relationship with the values measured. The detection and clustering thresholds and Kalman filter parameters were constant.

A client-server application was developed to store measurements and implement data fusion. This application also contains a presentation layer that allows the results to be presented using a web browser. Our application use a PostgreSQL database, Python standard packages, and the Django web framework.

## 3. Results

The fusion algorithm was tested using simulated data. Several numerical experiments were conducted to evaluate our system across multiple scenarios.

Two network topologies were considered. The first network $G_{1}\left(V,E\right)$ was a path graph. It consisted of linearly connected nodes (Figure 2).

The second network $G_{2}\left(V,E\right)$ (Figure 3) was a simplified version of the sample network available in the EPANET software. The original network was a water distribution network, so the edges were reversed to resemble a sewage network. The modification included transforming the acyclic graph into a tree via a depth-first search starting at node 1. The edge gains ($g_{e}$) and offsets ($o_{e}$) were calculated using pipe lengths ($l_{e}$) from the original EPANET network description (Equations (Equation 4) and (Equation 5)). The offsets were computed by dividing the pipe length by a constant velocity $v=10\frac{km}{h}$.
(4)$$o_{e}=\frac{l_{e}}{v}$$

The gains were calculated for each edge using a linear function. The minimal gain $g_{min}=0.65$ was chosen to make similar travel times for $G_{1}$ and $G_{2}$.
(5)$$g_{e}=\left[l_{e}-\min\left(l_{e}\right)\right]\cdot \frac{g_{min}-1}{\max\left(l_{e}\right)-\min\left(l_{e}\right)}+1$$

We simulated two types of sensors: (1) the microMole sensor system [6] and (2) liquid chromatography with tandem mass-spectrometry (LC-MS-MS). The microMole sensor system measures the pH and electrical conductivity (EC) of wastewater every second. It can be mounted in main sewer pipes of no less than 250 mm in diameter. The microMole system is not capable of identifying chemical compounds. LC-MS-MS is laboratory equipment capable of detecting and quantifying chemical compounds. Within the H2020 SYSTEM project [7], LC-MS-MS is used for analysis of wastewater samples collected at WWTPs. It analyses the composition of wastewater every 10 min. As LC-MS-MS is located at the WWTP, LC-MS-MS data are not sufficient for localizing the source of pollution in a sewage network graph.

Our sensors measured one of three entities of different characteristics:
$Q_{1}$with a range of [0,+∞) and a neutral value of 1400, which refers to the electrolytic conductivity,$Q_{2}$with a range of [0,14] and a neutral value of 7.65, which refers to the pH, and$Q_{3}$with a range of [0,1] and a neutral value of 0, which indicates the relative concentration of a pollutant.

The substances (compounds) that were tracked are listed in Table 1. The illegal substance is sodium hydroxide, described in Section 1.1. Pipe cleaner is legal but has a similar pH and electrolytic conductivity. The presence of those compounds in the proximity of the sensors measuring $Q_{1}$ and $Q_{2}$ affected readings in the same way: a positive peak of $Q_{1}$ ($Q_{1}$+) and a negative peak of $Q_{2}$ ($Q_{2}-$). The measured values of $Q_{3}$ were influenced only by $C_{2}$ in the form of a positive peak ($Q_{3}+$).

Thresholds for Algorithms 1, 2 and 4, and the Kalman filter parameters were constant for all simulations. The specific values were derived using the expectation-maximization algorithm on a representative sample of the measured values.

The average results of four experiments are presented below. For each network topology, the influence of sensor coverage, substance discharge amount, update period, and downstream propagation depth was calculated. The parameter values are presented in Table 2.

The updated period *T* was the constant time period used in resampling that determined how many iterations of the data fusion algorithm were performed.

The sensor coverage was represented by a number in the range [0,1], which expressed the number of sensors in the network relative to the number of nodes. For a given simulation, $Q_{1}$ and $Q_{2}$ sensors were placed randomly among all nodes (except the sink) using sampling without replacement. A single sensor measuring $Q_{3}$ was always located in the sink of the network.

### 3.1. Simulations

To create random scenarios for the data fusion module, a measurement generation module was produced. The method for creating simplified sensor observations was conceived based on the results of real-world experiments.

To create a simulation scenario, several parameters of the discharge event were required: the compound, node, amount, noise, and function inverse to the mapping function described in Section 2.2. An additional edge *e* parameter known as gain $g_{e}$ was also required. The edge gain $g_{e}$ was a real number that satisfied $g_{e}\in \left[0,1\right]$. The amplitude of the signal measured at the edge end divided by the amplitude of the signal measured at the edge start yielded $g_{e}$. The gain parameters revealed how the signal was attenuated while the compounds traveled through the edges. Noise was introduced by adding random values from a Gaussian distribution with a mean of 0 and a standard deviation equal to the product of the measurement value and the noise parameter.

Real-world measurements often resemble exponential functions with bases in the range from 0 to 1. In our experiments, a rectangle impulse function was used to simplify reverse mapping of the amount of the compound. Generating a single measurement series consisted of an initial calculation of the target area between the baseline reading and the measured values, and then the generation of a suitable number of measurements. A series corresponding to a single discharge event differed only in signal length (which was calculated by dividing the target area by the product of the gains of all the edges from the discharge node to the current node) and the initial signal amplitude (which was a property of the entity).

During one discharge event, many measurement series were generated that corresponded to each sensor in each node on the path from the discharge node to the sink. Scenarios in which more than one discharge event occurred were not taken into account, as this would have required knowledge of the behavior of compounds when they mix in the sewage network.

### 3.2. Quality of Data Fusion

For each scenario, a set of events was generated by the system using many iterations of the fusion algorithm depicted in Figure 1. Scenarios involved the simulated data of a single discharge in a random node, a node where pollution was introduced into the network was selected using a uniform distribution. The detected events were labeled either true positive or false positive. It should be noted that at most, one event was true positive.

Based on these labels, metrics were calculated for each simulation:The confidence coefficient, which was computed by dividing the confidence of the true positive event by the average confidence of all events. This metric showed how the confidences of true positive events compared to the confidences of false positive events. For the system to be useful, this metric had to be greater than 1.The number of reported events. The ground truth was 1. The smaller this number was, the more precise the localization. In studied scenarios, multiple events signified multiple possible nodes of discharge or multiple compounds; therefore, this was a valuable metric that demonstrated the precision of the system.

The results demonstrated that, as expected, the performance of the system depended on the sensor coverage of the network. According to Figure 4, the number of generated events decreased faster with an increase in the number of sensors in the network. In the case of the simple “path” network ($G_{1}$), the event count plot showed a median count of approximately 20 for a coverage of 10%. Taking into account that the a priori knowledge included two similar compounds, the system should reduce the source of the pollution to approximately 10 nodes. A coverage of 20% provides a twofold decrease in the event count.

According to simulations performed on the more realistic “Net2” network ($G_{2}$), the event count should not exceed 15 for coverage of 10% or greater. Moreover, taking into account that two similar compounds were considered, an event should be reduced to approximately seven nodes. Achieving such coverage in real networks may not be possible, but this metric provides a valuable overview of what can be expected concerning the performance of the system. It is important to note that to expect reconstructed events in a single node, sensor coverage would have to reach 100%, which in practice is impossible to achieve. This fact, however, does not mean that one cannot obtain accurate results from the proposed system at low sensor coverage. This means that the lower this value is, the more nodes must be considered as a potential source of pollution.

Figure 5 shows that if the position of the sensors and the discharge node are aligned in a way that allows for any detection (confidence coefficient ≠0), assuming that coverage is greater than 10%, it can be expected that true events have greater confidence than false events. Across all experiments with coverage of greater than 10%, true events had ≈30% greater confidence than false events in the simple network and ≈50% greater confidence in the more complicated network.

When it comes to correct identification of the source node, Figure 6 shows that we can get very close to 100% identification chance with network coverage of 80%.

Comparing these results with events counts shown in Figure 4, we can observe that even though a high coverage is needed to achieve excellent accuracy, results achieved at lower coverage would still be useful. When the number of sensors is low and the sensors are located far from the pollution source, we can expect that the system would generate several similar events in multiple nodes in proximity of the source. It is difficult to distinguish the actual source in such a case. However, as already mentioned (Figure 5), on average, our system assigns higher confidence values to actual pollution sources than to neighbor nodes.

Di Cristo and Leopardi in 2008 achieved a location identification rate from 60.5% to 100% with 31% of nodes containing sensors [30], while our system needed 60% coverage to get such high values. However, in the cited article, sensor locations were constant across simulations and only one discharge node was considered. In contrast, we considered random placement of sensors and the discharge node was chosen at random. Additionally, Di Cristo and Leopardi used hydraulic simulation by the EPANET simulator, which causes the performance of the system to be dependent on simulation quality. Our aim was to create a system which could continuously monitor the network and perform calculations as new measurements appear.

### 3.3. System Performance

The performance of the system was also evaluated by analyzing the algorithm execution time and the total number of observations that translated into the usage of system memory.

Figure 7 illustrates that memory usage (observation count) was directly proportional to the number of sensors. Analysis of the simulation run time charts (Figure 8) showed that the time complexity of the used algorithms was linear relative to the number of measurements that generated detections. As shown in Figure 8, the system can be expected to process more than 100 sensor observations per second.

## 4. Conclusions

The proposed localization strategy allowed the source of the pollution to be pinpointed to a small number of nodes in the networks. Our system correctly detected nearly 100% of events when sensors were present in at least 10% of the network nodes. To achieve such high location identification rates we needed 60% coverage. A large number of sensors were not required to be placed in the network to achieve meaningful results. The algorithms presented in this study can be expected to process at least 100 sensor observations per second.

Further research will focus on model formation for acyclic graphs, not only trees. This will allow us to consider all possible flow paths, therefore improving the quality of the results of data fusion. An additional benefit of this future approach will be the ability to use an unmodified network model in our system, thus removing the need for additional decisions regarding the modification process. Moreover, we will implement of parallel processing to increase the observation processing rate. The research will also include the use of machine learning algorithms in the event generation stage [45] to achieve improved quality compared to the algorithm based on threshold that is currently in use. Last, we plan to consider the uncertainty of measurements by incorporating this into the confidence coefficient of generated events.

## Figures and Tables

**Figure 1 sensors-21-00826-f001:**
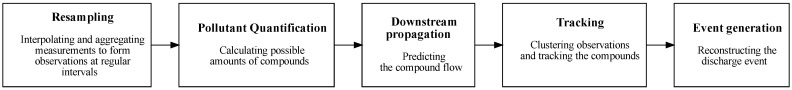
The data fusion algorithm.

**Figure 2 sensors-21-00826-f002:**
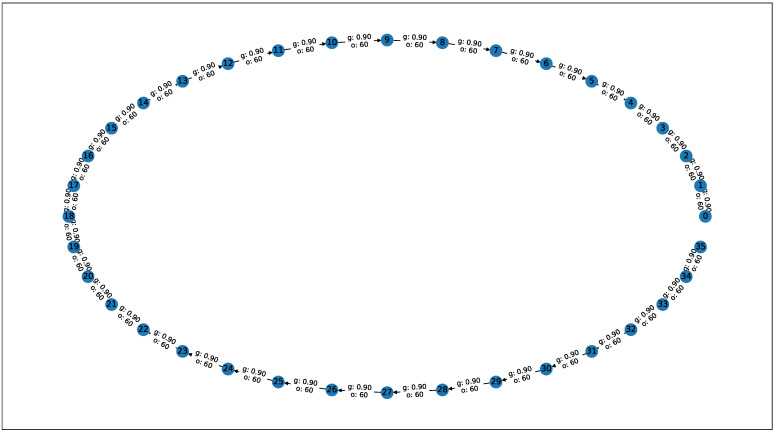
The $G_{1}\left(V,E\right)$ network topology used in simulations.

**Figure 3 sensors-21-00826-f003:**
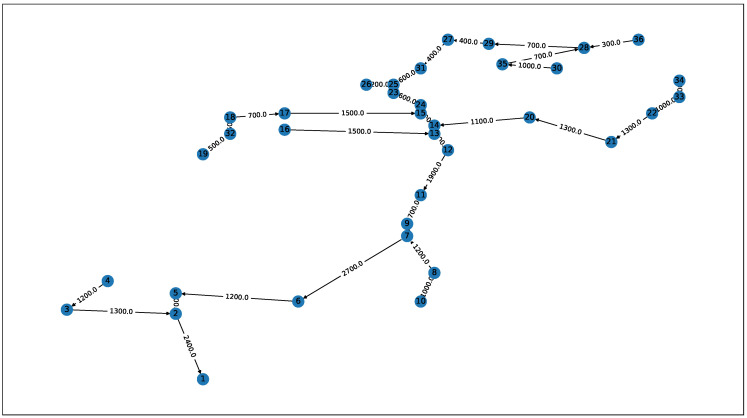
The $G_{2}\left(V,E\right)$ network topology used in simulations (based on the EPANET sample network 2).

**Figure 4 sensors-21-00826-f004:**
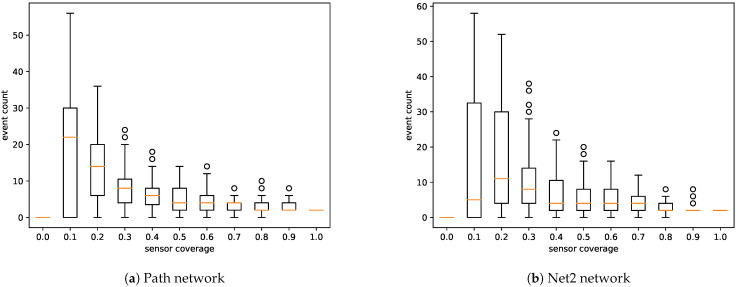
Event count by sensor coverage. The number of false positives rapidly decreased with an increase in sensor coverage.

**Figure 5 sensors-21-00826-f005:**
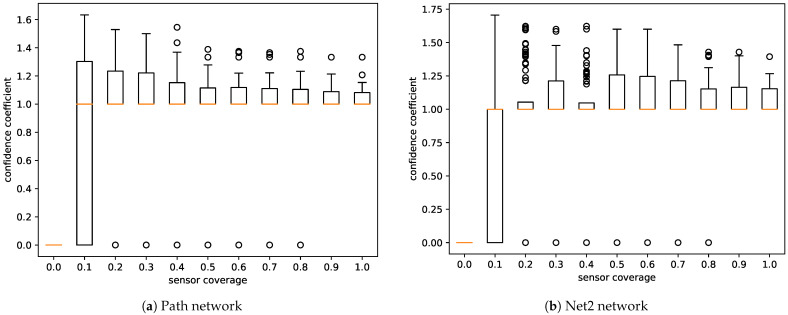
Confidence coefficient by sensor coverage. True positive events had confidence greater than the average confidence across all events.

**Figure 6 sensors-21-00826-f006:**
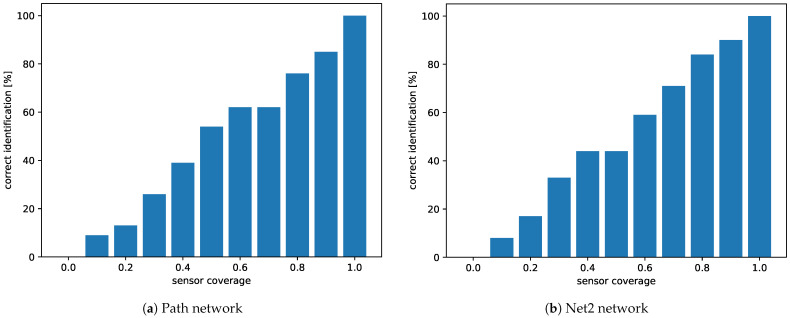
Percentages of identification of the right source node. Accuracy grows with an increase in sensor coverage.

**Figure 7 sensors-21-00826-f007:**
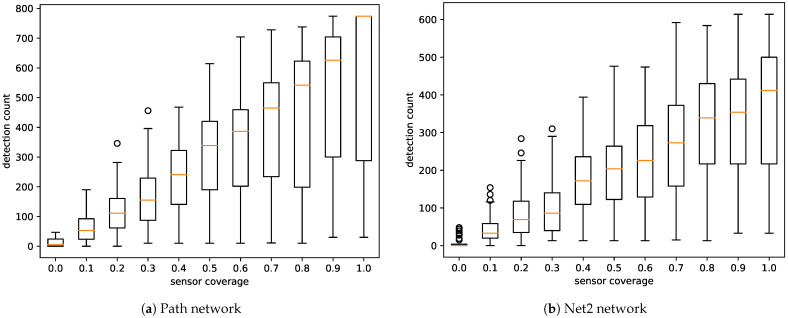
Detection count by sensor coverage. The number of observations increased linearly with coverage.

**Figure 8 sensors-21-00826-f008:**
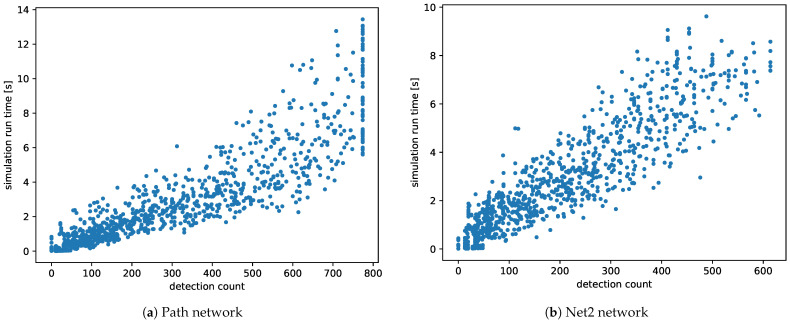
Calculation run-time by the number of detections. The time of execution was directly proportional to the total number of observations.

**Table 1 sensors-21-00826-t001:** Substances tracked by the data fusion system.

Short	Substance	Legality	pH	EC [mS/cm]
$C_{1}$	Pipe cleaner	Legal	12	22–26
$C_{2}$	Sodium hydroxide, NaOH	Illegal	12	1

**Table 2 sensors-21-00826-t002:** Parameters used in numerical experiments.

Parameter	Default Value	Considered Values
Update period [s]	1	1, 2, 3, 4, 5, 6, 7, 8, 9, 10
Sensor coverage	0.5	0, 0.1, 0.2, 0.3, 0.4, 0.5, 0.6, 0.7, 0.8, 0.9, 1
Discharge amount [l]	25	25, 50, 75, 100, 125, 150, 175, 200
Downstream propagation depth [nodes]	1	1, 2, 3

## Data Availability

Data sharing not applicable.

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
