# Peer review of "Pollution Source Localization in Wastewater Networks"

_sensors, 2021, doi:10.3390/s21030826_

Round 1

Reviewer 1 Report

the paper proposes a method to detect sources of pollution in utilities networks.

The method seem appropriate, but in the results it is not clear to what extent the causes of pollution were correctly found. Dealing with simulated data, this should be a must, that is, deciding where in the network put a source of pollution and check if the method reveals it. On the opposite in the result this is an issue that apparently was scarcely considered or, at least, the description is so obscure that one may not find this important issue. Moreover, an application to a real network to solve a real problem should be proposed.

The English is very poor and should be deeply revised.

Reviewer 2 Report

The article presents a contribution that could be very interesting if presented in a different way. The presentation of the algorithms is more or less well presented, however, the presentation of the results in simulation is very brief but tedious.

On the other hand, I do not think that it falls within the scope of the journal since the part corresponding to the sensors is treated very lightly in the article. It would have been interesting if authors delved a little more into the "Domain Switch" algorithm.

I suggest that the authors strengthen everything related to sensors and simulation analysis with an uncertainty analysis.

Reviewer 3 Report

The manuscript reports on the localization of pollution sources in utility networks through sensor observations.

The article is an original contribution and the topic is of interest for the readership of the Sensors journal.

English language is clear, and the presentation is adequate. The provided figures are necessary for the presentation and the understanding of the results. Anyway, I have detected some criticisms in the text that should be properly addressed.

The Authors can benefit from the comments below to improve their paper. These have to be accomplished before manuscript acceptance.

Abstract

The abstract is concise and reflects the content of the article.

Introduction

Aims of the study are clarified in the Introduction and supported by relevant references.

The introductory discussion concerns both the Water Distribution System (WDS) and Waste Water Network (WWN).

Lines 49-50: Concerning the research on design and planning methods that provide the most effective localization for a given set of sensors for detecting any anomaly event, the Authors should briefly mention the importance of the sectorization of WDNs. A proper division of a WDS into District Metered Areas provides important management benefits also with regard to implementation of monitoring, warning and emergency acting systems against accidental or intentional water contamination.

In this regard, the following references should be included as part of the discussion:

  • Di Nardo A., Di Natale M., Guida M., Musmarra D. (2013a) Water network protection from intentional contamination by sectorization. Water Resources Management 27:1837–1850. doi:10.1007/s11269-012-0133-y.
  • Ciaponi C., Murari E., Todeschini S. (2016). Modularity-Based Procedure for Partitioning Water Distribution Systems into Independent Districts. Water Resources Management, 30(6): 2021-2036, DOI: 10.1007/s11269-016-1266-1.

Methods

This section is clear and adequately detailed. The provided algorithms are useful for understanding the steps of the data fusion algorithm.

Line 117: Figure 3 is introduced in the text before Figure 1 and 2.

Line 129: Replace “even” with “event”. Typo error.

Line 157: In Figure 1 step 3 is indicated as “downstream propagation”, while the title of the third subsection is “Dilution backtracking”. According to the other subsections I suggest adopting the same designation as in Figure 1.

Results

This Section is presented in a logical sequence.

Lines 227-228 and caption of Figure 3: the Authors should briefly explain the choice of a (modified) WDS as an example of sewerage network. Otherwise, in my opinion, it is not relevant for the study the information that the second examined network was a simplified version of a sample network available in the EPANET software.

Line 234: Please, specify the acronym “EM”.

In Table 1: the units of the downstream propagation depth should be added.

Conclusions

Conclusions seem reasonable and are supported by the results.

Line 317: A model formation for acyclic graphs is a very important improvement of the model for a wider application of the localization strategy. I suggest the Authors to discuss more on this future research and on the expected results compared to the case of transforming the acyclic graph into a tree.

References

Several relevant references are included in the paper. Two references are suggested on the importance of sectorization of WDSs with regard to implementation of monitoring, warning and emergency acting systems against accidental or intentional water contamination. Apart from these, based on my knowledge, no important reference is missing.

Reviewer 4 Report

it is a good paper but no significant indicators of performance are used. Include references connected to MDPI journals, such as Remote Sensors, Sensors, etc. What is beyond the present localization methods  in terms of comparison? it means, please use another common approach as comparison.

1) Formula number should be provided in the manuscript.

2) Only the EPANET Example can not show the performance of the present method in real-world running water monitoring system. Please add your own engineering application cases.

Round 2

Reviewer 2 Report

I thank the authors for the improvements they made to the article. It is now in publishable form.

Reviewer 3 Report

The manuscript has been significantly improved following the recommendations of the Reviewers; all my concerns have been addressed and convincingly justified.

Note that at lines 70-73: references [28] and [29] must be interchanged; Ciaponi et al. in [28] proposed a methodology that combines an algorithm for the automated creation of district metered area (DMA) boundaries with practical criteria for DMA design. Di Nardo et al. in [29] focused on proving the benefits of partitioning by simulating a discharge of cyanide and investigating the influence of district isolation on the security of a water supply system.

Reviewer 4 Report

Good revision to be accepted.